# CYP3A Excipient-Based Microemulsion Prolongs the Effect of Magnolol on Ischemia Stroke Rats

**DOI:** 10.3390/pharmaceutics12080737

**Published:** 2020-08-05

**Authors:** Jiun-Wen Guo, Chih-Cheng Chien, Jiann-Hwa Chen

**Affiliations:** 1Department of Medical Research, Cathay General Hospital, Taipei 10630, Taiwan or cgh393831@cgh.org.tw; 2Ph.D. Program in Pharmaceutical Biotechnology, College of Medicine, Fu Jen Catholic University, New Taipei City 24205, Taiwan; 3Department of Anesthesiology, Cathay General Hospital, Taipei 10630, Taiwan; chienc@cgh.org.tw; 4School of Medicine, College of Medicine, Fu Jen Catholic University, New Taipei City 24205, Taiwan; 5Department of Emergency Medicine, Cathay General Hospital, Taipei 10630, Taiwan

**Keywords:** magnolol, microemulsion, neuroprotection effect, brain ischemic, CYP3A

## Abstract

Magnolol, which is a CYP3A substrate, is a well-known agent that can facilitate neuroprotection and reduce ischemic brain damage. However, a well-controlled release formulation is needed for the effective delivery of magnolol due to its poor water solubility. In this study, we have developed a formulation for a CYP3A-excipient microemulsion, which can be administrated intraperitoneally to increase the solubility and bioavailability of magnolol and increase its neuroprotective effect against ischemic brain injury. The results showed a significant improvement in the area under the plotted curve of drug concentration versus time curve (AUC0–t) and mean residence time (MRT) of magnolol in microemulsion compared to when it was dissolved in dimethyl sulfoxide (DMSO). Both magnolol in DMSO and microemulsion, administrated after the onset of ischemia, showed a reduced visual brain infarct size. As such, this demonstrates a therapeutic effect on ischemic brain injury caused by occlusion, however it is important to note that a pharmacological effect cannot be concluded by this study. Ultimately, our study suggests that the excipient inhibitor-based microemulsion formulation could be a promising concept for the substrate drugs of CYP3A.

## 1. Introduction

Magnolol (5,5′-diallyi-2,2′-dihydroxybiphenyl, C18H18O2, MW = 266.33) and honokiol, which are isomers of hydroxylated biphenolic compounds, are the primary active agents that have been isolated from the herb *Magnolia officinalis* [1,2]. Magnolol has been widely used in several treatments, as an anxiolytic [3,4], analgesic [5], antidepressant [6], antimicrobial [7], antispasmodic [8], anti-tumorigenic [9], and for anti-Alzheimer’s therapy [10]. A recent study also indicated that oral magnolol could prevent memory loss in a model animal by preserving cholinergic neurons [11]. In addition, our previous studies have demonstrated that magnolol can enhance neuroprotection by down-regulating p38/MAPK, CHOP, and nitrotyrosine, subsequently reducing brain damage caused by ischemic stroke [12,13].

A study in rats showed that CYP3A4 was the major subtype responsible for further hydrolytic metabolism of magnolol and honokiol [14]. In addition, magnolol and honokiol are relatively strong inhibitors of rat CYP1A2 and CYP2E1, but are weak inhibitors of rat CYP3A and human CYP3A4 in vitro [15]. Therefore, we can consider magnolol and honokiol to be the substrates of CYP3A4. Previous studies demonstrated that Tween-80 and Cremophor RH40 inhibited the CYP3A activity in hepatocytes and liver microsomes [16,17]. Rao et al. also reported that Cremophor RH40 or Tween 80 were two potential excipients based in the self-microemulsiflying formulation that could reduce the metabolism of CYP3A substrates [18]. Thus, a CYP3A-excipient base formulation design may improve the bioavailability of magnolol (4%) [19].

Due to the poor water solubility of magnolol (1.24 mg/L at 25 °C,) [20], dimethyl sulfoxide (DMSO, approximately 16 mg/mL, Item No. 14233, Cyman Chemical, Ann Arbor, MI, USA) is often used as a solvent [12] or is dispersed in a saline suspension [11] for magnolol delivery because of its ability to dissolve polar and nonpolar substances. Although the physiological and pharmacological effects of DMSO are not fully understood, it is known to have a toxic effect such as causing the release of histamine from mast cells. Therefore, caution is applied when DMSO is implemented as a vehicle for other drugs [21]. Despite recent studies showing that the systemic toxicity of DMSO is low, there has only been one DMSO-containing topical formulation that has obtained Food and Drug Administration (FDA) approval in the United States so far. In addition, DMSO is yet to be approved for formulation to allow for oral or other delivery systems [22]. However, Neoral is an FDA-approved cyclosporine microemulsion formulation for oral use and Restasis is a topical application for ophthalmic use [23]. Therefore, we can assume that it is safer to use a microemulsion for clinical use as a drug delivery vehicle than DMSO even though the long-term use of DMSO as a solvent for drug delivery remains a safety concern. Thus, a well-controlled release formulation is necessary for the effective delivery of magnolol. In this study, we developed a CYP3A-excipient-based microemulsion formulation to improve the solubility and bioavailability of magnolol to increase its neuroprotective effect against ischemic brain injury. The findings of this study inform the future use of magnolol in treatments and the potential utility of an excipient-based microemulsion formulation as a substrate of CYP3A drugs.

## 2. Materials and Methods

### 2.1. Materials

Magnolol, Honokiol, Oleic acid (OA), 1,2-propylene glycol (1,2-PG), Triglyceride (TG), Tween 80, and Polyethylene glycol (PEG 400) were purchased from Merck KGaA (Darmstadt, Germany). All other chemicals used were of analytical grade.

### 2.2. Construction of Pseudo-Ternary Phase Diagrams

For the determination of the existing zone of microemulsion (ME), pseudo-ternary phase diagrams were constructed using the aqueous titration method. The selected oil phase (TG: OA, 3:1) was mixed with surfactant: the cosurfactant mixture was 1:1:1 of Tween 80, 1,2-PG, and PEG 400. The ratios of the mix used for titration were 1:9, 2:8, 3:7, 4:6, 5:5, 6:4, 7:3, 8:2, and 9:1. The mixture was then titrated with distilled water until it became turbid, and the volume of water used was recorded.

### 2.3. Characterization of Microemulsion

#### 2.3.1. Measurement of Droplet Size and Zeta Potential of Microemulsion

Droplet size and zeta potential were measured using an SZ-100 Nanoparticle Analyzer (Horiba Ltd., Kyoto, Japan). All the measurements were recorded and conducted under ambient conditions at room temperature (25 °C ± 1 °C) and in three independent samples.

#### 2.3.2. Viscosity of Microemulsion

The viscosity of the formulation was measured using a Visco-895 rotational viscometer (Atago Co., Inc., Tokyo, Japan) using the spindle A3 RE-77106 at room temperature (25 °C ± 1 °C) in three independent samples.

#### 2.3.3. Determination of the Maximum Solubility of Magnolol in ME

Solubility studies were conducted as described by Thomas et al. [24], with minor modification. A volume of 1 mL of ME was added to a small vial and the excess amount of magnolol was added. The vial was tightly stoppered and continuously stirred for 72 h in a mechanical shaker at room temperature (25 °C ± 1 °C), then centrifuged at 13,200 rpm for 10 min. The supernatant was then separated, filtered, and appropriately diluted with methanol, and the magnolol content was quantified using high-performance liquid chromatography (HPLC).

#### 2.3.4. Stability Studies

The optimized ME was stored at room temperature (25 °C ± 1 °C) for one month. Then, the consistency and maximum concentration of magnolol was investigated at 0, 14, and 30 days. The samples were appropriately diluted with methanol and the magnolol content was quantified using HPLC.

#### 2.3.5. Serum Stability Study

An amount of 100 μL ME was incubated with the equivalent rat serum (1:1) for 12 h at 37 °C. The droplet size was measured using an SZ-100 Nanoparticle Analyzer (Horiba Ltd., Kyoto, Japan) at 1, 3, 6, and 12 h [25].

#### 2.3.6. Transmission Electron Microscopy (TEM)

A drop of ME was applied to a 200-mesh carbon-coated copper grid and stained with 2% of phosphotungstic acid. After drying, the samples were then examined under the JEM-2000EX II (JEOL Ltd., Tokyo, Japan) instrument operating at 100 kV. TEM micrographs were recorded using image analysis software and a CCD camera (Orius, Gatan Inc., Pleasanton, CA, USA) [26].

### 2.4. Animals

A total of 33 male Sprague-Dawley (SD) rats (BioLASCO Taiwan Co., Ltd., Taipei, Taiwan), weighing 200 ± 20 g, were housed under conditions of controlled humidity (40%) and temperature (22 °C ± 2 °C), with a 12-h light–dark cycle. All animal experiments were conducted in accordance with accepted standards of humane animal care under protocols approved by the Institutional Animal Care and Use Committee of Cathay General Hospital (106-014, 22 November 2017).

### 2.5. Blood and Brain Sample Collection for Pharmacokinetic Study

The SD rats were administrated 30 mg/kg of 10 mg/mL magnolol in ME or in dimethyl sulphoxide (DMSO) both intraperitoneally and orally [12]. For safety and toxicity issues, rats were not orally administrated magnolol in DMSO. Blood samples (100 μL) were collected from the rats’ tail vein into heparinized tubes at various time points: 5, 10, 15, 30, 45, and 60 min and 2, 4, 6, and 24 h post-treatment. There were three rats in each treatment (magnolol in DMSO or ME) at each blood sampling time point. Brain samples were collected at 1 and 24 h post-treatment (n = 3 for each time point per treatment). Brain sections were obtained and weighed. The brain tissues were mixed with 0.6 mL acetonitrile to denature proteins [27]. The homogenate, which contained 0.1mg/mL honokiol, as an internal standard, was centrifuged at 13,200 rpm for 10 min at room temperature (25 °C ± 1 °C). Serum was separated by centrifugation at 4500 rpm for 3 min at 4 °C, and then equal volumes of acetonitrile with 0.1 mg/mL honokiol, as the internal standard, were added to denature proteins. The supernatant and 20 μL serum samples were then submitted to HPLC analysis.

### 2.6. High-Performance Liquid Chromatography System and Method Validation

The HPLC analysis was performed using a Primaide 1110 pump, Primaide 1410 UV detector, a Primaide 1210 auto-sampler (Hitachi, Tokyo, Japan), and a mightysil RP-18 column, 4.6 × 250 mm, 5 μm (Kanto chemical Co. Inc., Tokyo, Japan). The mobile phase was methanol–water (80:20, *v*/*v*, pH 2.5–3 adjusted by oethophosphoric acid), filtered through a 0.45 μm millipore filter. The flow rate was 1 mL/min, and the sample injection volume was 20 μL. Detection was performed at a wavelength of 292 nm for magnolol and honokiol at room temperature (25 °C ± 1 °C) with retention times 7.1 ± 0.2 min and 9.9 ± 0.3 min in honokiol and magnolol, respectively [28]. The assay used power regression in the concentration range of 0.02–20 μg/mL for magnolol. The inter- and intra-day assay accuracy (% error) and precision (% coefficient of variation (CV)) was between −6.7 and 4.2%, and 0.7 and 5.6%, respectively, for magnolol at a concentration of 0.02 μg/mL (Appendix A).

### 2.7. Calculation of Pharmacokinetic Parameters

Pharmacokinetic parameters were calculated using the one-compartment model for the intraperitoneal administration study and two-compartment model for the oral administration study by a free Microsoft excel add-in program, pksolver [29]. The peak serum concentration (Cmax) and time to peak concentration (Tmax) were calculated based on experimental measurements. The area under the plot of drug concentration versus time curve (AUC0-t) was calculated using the trapezoidal rule.

### 2.8. Surgical Procedures for Inducing a Reversible Ischemic Stroke

A reversible, temporary ischemia was induced in the SD rats through occlusion of the left middle cerebral arteries (MCAs) [30]. The left common carotid arteries (CCA) were ligated with clips, under anesthesia, using 2–4% isoflurane in oxygen via a facemask. A silicone-coated 4-0 nylon monofilament (RWD Life Science Co., Ltd., Shenzhen, China) was introduced into the CCA and was gently advanced through the internal carotid artery until its tip occluded the origin of the MCA [31,32]. The 4-0 nylon monofilament was withdrawn 90 min after the onset of ischemia.

#### 2.8.1. Behavioral Tests

The behavioral tests included the elevated body swing test (EBST) and the grip strength assessment.

##### EBST

During the recovery period of MCA occlusion-induced ischemic stroke, each rat was suspended above a table by its tail. While the MCA occlusion (MCAO)-induced ischemic stroke successfully damaged the left focal cerebral, a biased swing to the right side demonstrating a “C-shaped lateral bending body” was observed immediately (Appendix A) [33,34].

##### Grip Strength Assessment

MCAO-induced ischemic stroke rats were evaluated using the gripping strength meter (Bio-Cando Biotechnology Inc., Tao-Yuan, Taiwan). Rats were placed on the string midway between the supports and pulled back quickly in the horizontal direction when their paws grabbed the bar. Forelimb griping strength was recorded when the grip was released. Grip strengths of forelimbs were recorded at 90 min and 24 h after MCAO and before the operation as the normal health baseline [35,36].

### 2.9. Magnolol Administration for Pharmacodynamic Study

After the successful induction of an ischemic stroke, a total of 18 SD rats were randomly assigned to three groups: (a) stroke group without treatment, (b) magnolol in the DMSO group, and (c) magnolol in the ME group. A quantity of 30 mg/kg magnolol in DMSO or in ME was administrated intraperitoneally 90 min after the onset of ischemia as our previous study showed that 30 mg/kg magnolol had a pharmacodynamic effect [12].

#### Evaluation of the Size of the Ischemic Injury Using 2,3,5-triphenyl tetrazolium Chloride Staining

The brains of the rats were cut into five 2-mm coronal slices using a rat brain matrix (Stoelting Co, Wood Dale, IL, USA). Sections were then incubated in 2% 2,3,5-triphenyl tetrazolium chloride (TTC) for 30 min at room temperature (25 °C ± 1 °C) and then immediately fixed in 10% formaldehyde overnight at 4 °C. The infarcted area was outlined in white and measured using the ImageJ 1.48 software (Nation Institutes of Health, USA) on the posterior surface of each section. The mean volume of the brain infarction was calculated by summing the infarct area in each slice multiplied by the thickness of the slice to then represent the infarction area as a percentage of the entire brain volume [37].

### 2.10. Statistical Analysis

Data were presented as the mean ± SD. Student *t*-tests were performed using the SigmaPlot 10.0 software. For the infarct measures on the TTC sections study, one-way ANOVA followed by the Bonferroni post hoc test was performed by the IBM SPSS statistics software Version 20. *p* < 0.05 was considered statistically significant.

## 3. Results

### 3.1. Physicochemical Properties of Selected Microemulsions

The pseudo-ternary phase diagram of the microemulsion consisted of oil, surfactant, and water. Formulations A and B represented the chosen formulation (Figure 1). The droplet sizes and viscosity for formulations A and B were 111.2 ± 27.3 nm, 697.7 ± 208.3 nm, 21.86 ± 1.31 cP, and 155.72 ± 2.28 cP at room temperature (25 °C ± 1 °C), respectively (Table 1). However, formulation A was cloudy at 4 °C as a semi-microemulsion. Therefore, formulation B was chosen for further studies, and the maximum solubility of magnolol in formulation B was 22.78 ± 0.43 mg/mL. The stability of the maximum concentration of magnolol in formulation B was 23.33 ± 0.73 and 23.35 ± 0.35 mg/mL at 14 and 30 days post-treatment, respectively (Table 2). The serum stability study result also showed that formulation B had a slightly reduced droplet size from 663.1 ± 74.4 at 1 h to 553.1 ± 84.4 nm at 6 h with further reduction to 409.1 ± 73.3 nm (*p* < 0.05, compared to 1 h) at 12 h (Figure 2). This result suggested the relative stability of formulation B during blood circulation was for at least 6 h. The shape of formulation B showed various appearances under the observation by TEM (Figure 3). The particle diameter of formulation B observed by TEM is either smaller than or equal to the hydrodynamic diameter measured by the Nanoparticle Analyzer.

### 3.2. Pharmacokinetic Parameters of Magnolol Loaded ME

The mean serum concentration–time profiles of intraperitoneal treatment of magnolol in ME and DMSO showed that the Tmax of magnolol in ME was about four times slower than that in DMSO (Figure 4A,B; Table 3). However, compared to magnolol in DMSO, the AUC0–t and MRT of magnolol in ME were higher by 64% and 54% (*p* < 0.05), respectively (Table 3). In addition, the clearance (Cl/F) of magnolol in ME was 38% slower (*p* < 0.05) than in DMSO via intraperitoneal injection (Table 3). The mean serum concentration–time profiles of oral administration of magnolol in ME are shown in Figure 4C and the estimated pharmacokinetic parameters using the two-compartment model is listed in Table 3. The orally administrated magnolol in the ME group showed about 30% faster Tmax, about three times higher AUMC, and about ten times MRT compared to the intraperitoneal treatment of magnolol in the ME group.

### 3.3. Brain Distribution and Effects of Magnolol on an Induced Ischemic Stroke

Brain concentrations after 1 and 24 h of intraperitoneal treatment with 30 mg/kg magnolol showed that (Figure 5) the brain concentrations were similar after 1hr injection (*p* = 0.99) between two groups. However, magnolol in ME showed a relatively higher distribution (*p* = 0.048) as compared with magnolol in DMSO after 24h of being administrated intraperitoneally. The behavioral studies also showed that all animals demonstrated that C-shaped lateral bending body and grip strength dramatically decreased more than 50% compared to the baseline after MCA occlusion (MCAO) surgery (Figure 6). Both treatments of magnolol in DMSO and ME administrated after the onset of ischemia showed a protective effect on brain infarction size, as measured by TTC staining, compared to the stoke control (Figure 7, both *p* < 0.05). There was no significant difference between the total brain infarction size of magnolol in ME and DMSO. Furthermore, the grip strength and EBST showed no difference in all three groups.

## 4. Discussion

Magnolol exhibited strong radical scavenging, antioxidant, and anti-inflammatory activity that could significantly reduce the accumulation of superoxide anions, oxidative damage, apoptosis, and autophagy in the ischemic brain [38,39,40]. In this study, we developed a CYP3A-excipient ME, which was designed to be administrated intraperitoneally to increase the solubility and bioavailability of magnolol and increase its neuroprotective effect against ischemic brain injury. Magnolol in ME results in similar brain distribution after 1h of injection but a significantly higher concentration after 24h administration and superior protection; however, it is not significant compared with magnolol in DMSO. This phenomenon may have resulted from the delayed Tmax of magnolol in ME, which led to magnolol distribution delay to the ischemic brain. However, compared with magnolol in DMSO, magnolol in ME has a similar Cmax and brain concentration an hour after injection. A previous study showed that concentrations in various brain regions of magnolol are similar [27]. Thus, magnolol in ME has a slower but more sufficient magnolol distribution to the ischemic brain, thus achieving the neuroprotective effect. Our TTC stain results also confirmed that magnolol in ME provides sufficient magnolol to achieve a protective effect. However, the behavioral tests showed no significant difference in all three groups. This may be attributed to the primary MCAO damage that led to the death of dopaminergic neurons in substantia nigra [41,42]. Magnolol played a role in the antioxidant and anti-inflammation effects, for example, during the second reperfusion damage period [38,39]. There were no shreds of evidence that magnolol can enhance the regeneration of dopaminergic neurons in the brain motor area.

The intraluminal suture occlusion of MCA is the well-used method in ischemic stroke animal models and laser Doppler flowmetry can be used to ascertain the magnitude of occlusion or reperfusion and to assure the homogeneity of the infarcts [43]. A previous study [44] demonstrated that the percentages of infarction volume were around 30 to 40% in a 70% decrease in regional cerebral blood flow in the MCA territory monitoring by laser Doppler flowmetry. The behavior score, including body symmetry, gait, and circling behavior, showed a positive correlation with the percentages of infarction volume. Our TTC stain shows that about 35% of brain infarction size in both the stroke group and behavioral studies demonstrates that the C-shaped lateral bending body and grip strength dramatically decreased after MCAO surgery in all animals. Our TTC staining and behavior results are consistent with the previous study.

Our findings show that the CYP3A-excipient inhibitor-based microemulsion formulation that we have developed increases the AUMC and MRT as well as causes a larger tissue distribution amount of magnolol via oral administration compared to intraperitoneal treatment. This finding corresponds with previous studies that have demonstrated that tween 80 and cremophor RH40 successfully worked as CYP3A substrates in rat and human liver microsomes [17,18]. The excipient of CYP3A4 inhibitor-based formulations, such as tween 80 and cremophor RH40, is a potential protective principle for reducing the metabolism of CYP3A substrates [18,45]. Our pharmacokinetic findings regarding the slower Tmax and Cl/F (clearance) of magnolol in ME compared to DMSO, but larger AUC0–t and longer MRT of magnolol in ME, could indicate that the microemulsion controls the release of magnolol, consequently slowing absorption. Moreover, tween-80, the excipient of the CYP3A4 inhibitor, most likely reduces the clearance of magnolol and then raises the MRT, AUC, and brain distribution after 24 h of administration. Thus, the increased availability of magnolol in ME may lead to superior ischemic brain protection.

The strategy of blood-to-brain drug delivery involves improving BBB permeability of the drug–carrier conjugate. Nanocarriers such as micelles, nanoemulsions, and microemulsions are promising carrier vehicles for direct drug transport across the intact BBB. Due to the nano size, the drug–carrier conjugate can quickly enter the brain capillary endothelial cells by endocytosis or transcytosis, for example [46]. The selected formulation B is a clear solution that is entrapped in oil, water, and surfactants. Its droplet size and zeta potential, an essential index of suspension stability, are around 700 nm and −15 mV, respectively. A greater absolute zeta potential corresponds with higher stability of colloidal dispersions [47]. Our serum stability results also confirmed that formulation B can be relatively stable during blood circulation for at least 6 h. Electronic conductivity can define the nature of the microemulsion, with a previous study reporting microemulsion with a 10.3 to 52.5 μS/cm conductivity range, as a bicontinuous microemulsion with its microstructures formed in a water and oil continuous phase (bicontinuous) structure [48,49]. Since the conductivity of formulation B is about 16 μS/cm, it can be characterized as a bicontinuous microemulsion. Thus, formulation B presents various appearances under TEM observation. We also observed a relatively small particle size (about 50 nm, Appendix A) of formulation B. Furthermore, the amphiphilic nature of the bicontinuous microemulsion [49] as well as the relatively small particle micelle can simultaneously increase the solubility and permeability of magnolol for delivery from blood to brain.

## 5. Conclusions

As far as we know, this is the first record of utilizing a CYP3A-excipient microemulsion as a delivery system loaded with magnolol for treatments of ischemia stroke in rats. Our results have demonstrated improvements in a reduced visual brain infarct size as a result of magnolol loaded in a microemulsion showing a therapeutic effect on ischemic brain injury caused by occlusion, although a pharmacological effect cannot be concluded. Therefore, the excipient-based microemulsion formulation tested in this study is an appropriate substrate for CYP3A drugs and an optimal delivery carrier relative to DMSO for in vivo study. However, further research is required to re-formulate the compositions of the microemulsion to enhance the distribution of magnolol to tissues and the therapeutic outcome, as well as to clarify the pharmacological effects of magnolol on ischemic stroke.

## Figures and Tables

**Figure 1 pharmaceutics-12-00737-f001:**
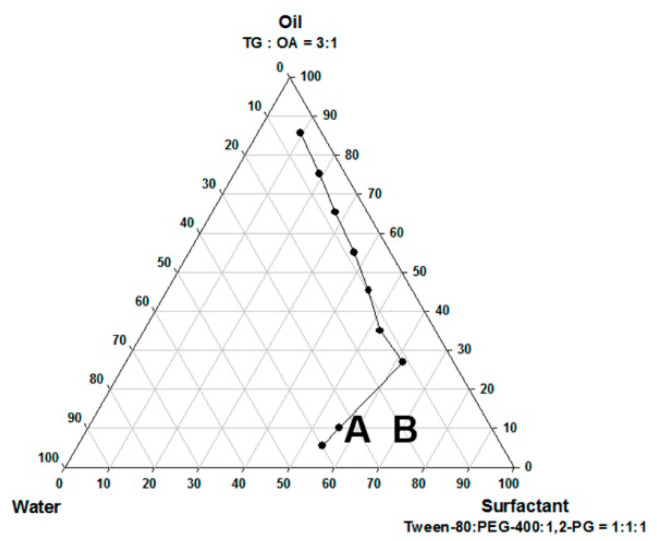
The pseudo-ternary phase diagrams of the microemulsion composed of oils, surfactants, and water. Formulations A and B represent the chosen formulation.

**Figure 2 pharmaceutics-12-00737-f002:**
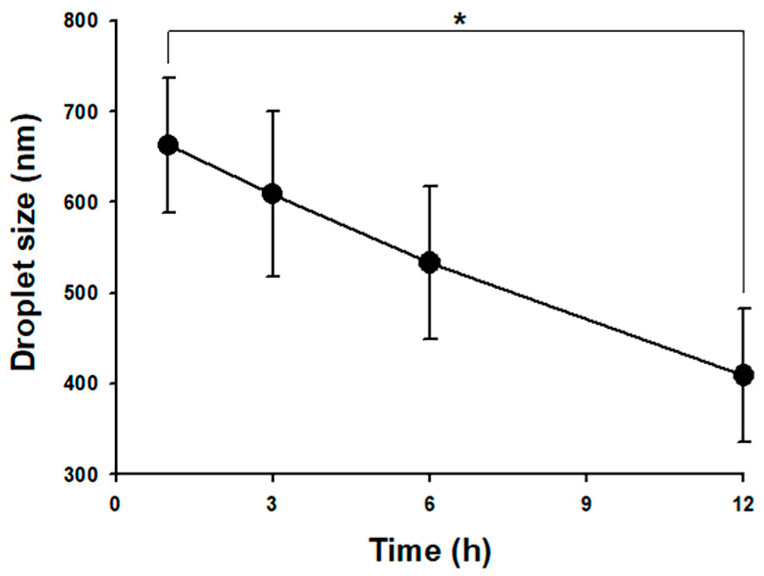
Serum stability of formulation B. Formulation B slightly reduced its droplet size and can be stable for at least 6h in serum. * *p* < 0.05. Mean ± SD, *n* = 3.

**Figure 3 pharmaceutics-12-00737-f003:**
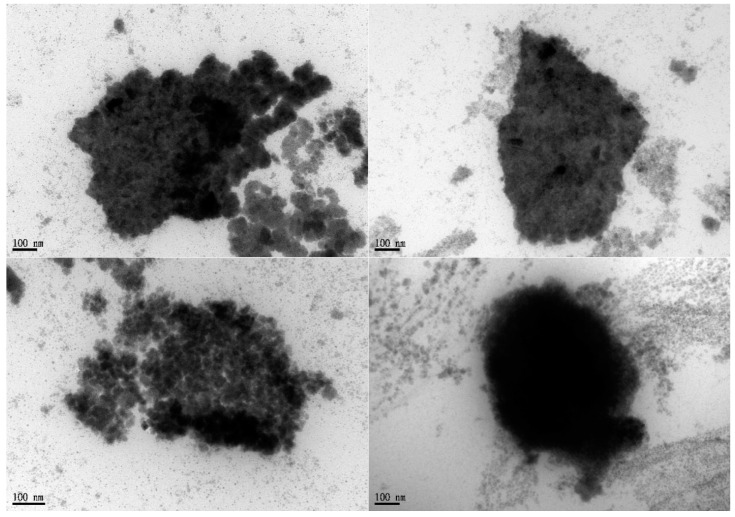
Formulation B represented in various appearances under the observation by TEM. The composition of formulation B was described in Table 1. TEM, transmission electron microscopy; Scale bar = 100nm.

**Figure 4 pharmaceutics-12-00737-f004:**
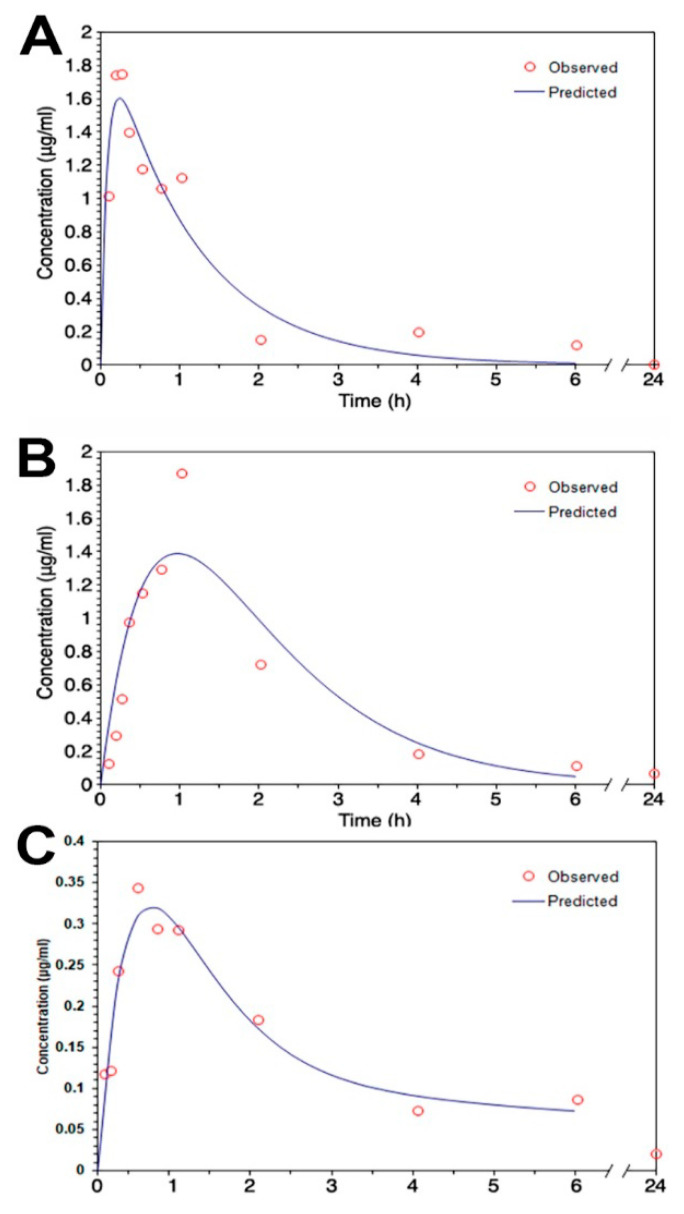
The mean serum concentration–time profiles of magnolol (**A**) in dimethyl sulfoxide (DMSO) and (**B**) in a microemulsion after 30 mg/kg of magnolol was administrated intraperitoneally (**C**) in a microemulsion after 30 mg/kg of magnolol was administrated orally. (*n* = 3).

**Figure 5 pharmaceutics-12-00737-f005:**
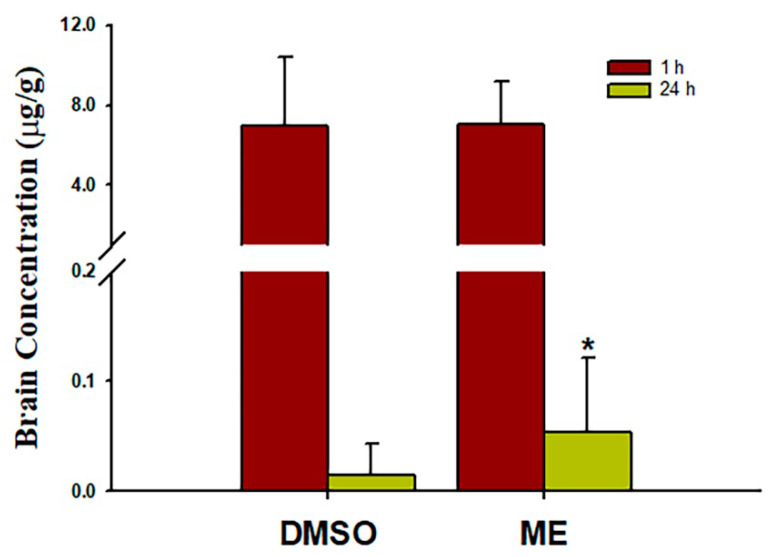
Mean brain magnolol concentrations in rats 1 and 24 h post-treatment with 30 mg/kg magnolol administrated intraperitoneally. Data are expressed as mean ± SD of three independent experiments. **p* < 0.05. ME, microemulsion group.

**Figure 6 pharmaceutics-12-00737-f006:**
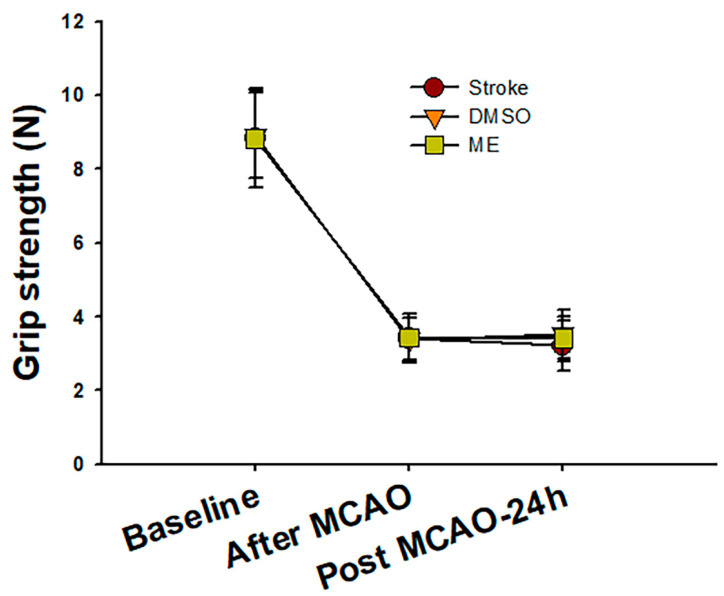
The change in rat grip strength during the experimental period. Rat grip strength dramatically decreased more than 50% compared to the baseline in all three groups after middle cerebral artery occlusion (MCAO) surgery. After 24 h of magnolol (30 mg/kg) treatment, the DMSO and ME groups showed a slight but non-significant grip strength increase compared to the stroke group. Data are expressed as mean ± SD of six independent experiments.

**Figure 7 pharmaceutics-12-00737-f007:**
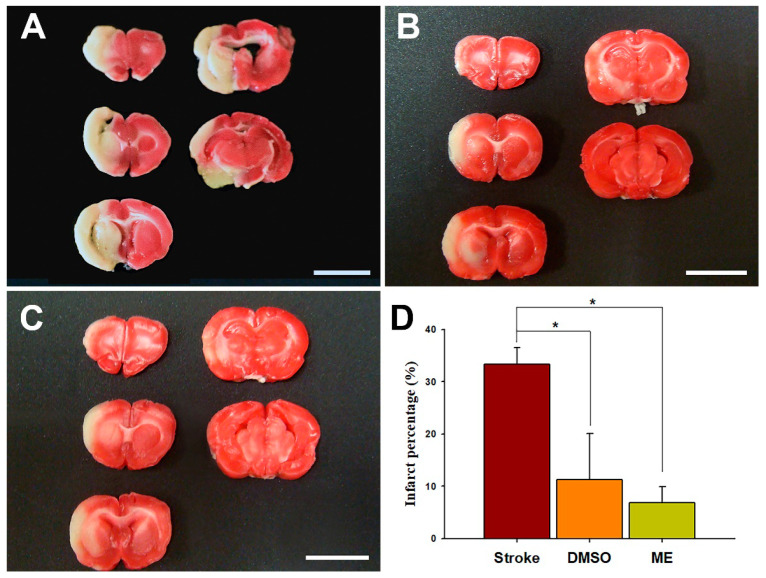
Effects of magnolol in DMSO and in ME on the brain infarct volume using an ischemic rat model. Magnolol was intraperitoneally administrated at 30 mg/kg 90 min after onset of ischemia. Brain infarction size was assessed by TTC staining present in 2 mm thick coronal sections from the rat brains after 24 h of ischemia. (**A**) Stroke group without treatment, (**B**) magnolol in the DMSO group, (**C**) magnolol in the ME group, and (**D**) mean percentage of brain infarct area. Scale bare = 1 cm. Data are expressed as mean ± SD of six independent experiments. * *p* < 0.05; one-way ANOVA (Bonferroni). ME, microemulsion group.

**Table 1 pharmaceutics-12-00737-t001:** The physicochemical properties of the two selected microemulsions. (Mean ± SD, n = 3).

Properties (Unit)	Formulation A	Formulation B
Composition of S:O:W	6:1:3	7:1:2
Droplet size (nm)	111.2 ± 27.3	697.7 ± 208.3
Size distribution (PI)	0.434 ± 0.007	0.319 ± 0.036
Zeta potential (mV)	-	15.51 ± 0.24
Viscosity (cP)	21.86 ± 1.31	155.72 ± 2.28
Electronic conductivity (µS/cm)	-	16.23 ± 0.45

S, surfactant (Tween 80: PEG 400: 1,2-PG = 1:1:1); O, oil (TG: OA, 3:1); W, water; -, un-tested.

**Table 2 pharmaceutics-12-00737-t002:** The stability of selected microemulsion, formulation B loaded with magnolol. (Mean ± SD, *n* = 3).

Solubility	Day 0	Day 14	Day 30
mg/mL	22.78 ± 0.43	23.33 ± 0.73	23.35 ± 0.35
maximum percentage (%)	91.3 ± 4.4	93.4 ± 2.9	93.3 ± 1.4

**Table 3 pharmaceutics-12-00737-t003:** Estimated pharmacokinetic parameters using a one-compartment model when a dose of 30 mg/kg of magnolol in the microemulsion or in dimethyl sulfoxide (DMSO) was administrated intraperitoneally. For the oral administration study, a two-compartment model was used to estimate the pharmacokinetic parameters (Mean ± SD, *n* = 3).

Parameter (Unit)	DMSO	-	Formulation B		-
Administration	Intraperitoneal	Intraperitoneal	*p* Value	Oral	*p* Value
t_1/2ka_ (h)	0.06 ± 0.01	0.66 ± 0.01 **	P^1^ < 0.01	0.31 ± 0.10	P^2^ < 0.01
t_1/2k10_ (h)	0.80 ± 0.20	0.69 ± 0.01			
t_1/2Alpha_ (h)				0.96 ± 1.00	
t_1/2Beta_ (h)				13.48 ± 4.14	
Tmax (h)	0.24 ± 0.03	0.97 ± 0.01 **	P^1^ < 0.01	0.66 ± 0.15	P^2^ < 0.05
Cmax (μg/mL)	1.61 ± 0.32	1.39 ± 0.27		0.32 ± 0.11	P^2^ < 0.01
AUC0-t (μg/mL·h)	2.23 ± 0.26	3.66 ± 0.75 *	P^1^ < 0.05	1.50 ± 0.41	P^2^ < 0.05
AUMC (μg/mL·h^2^)	2.80 ± 0.77	7.10 ± 1.53 *	P^1^ < 0.05	23.18 ± 4.06	P^2^ < 0.01
MRT (h)	1.25 ± 0.30	1.93 ± 0.02 *	P^1^ < 0.05	13.02 ± 1.14	P^2^ < 0.01
Cl/F (mg/kg)/(μg/mL)/h	13.54 ± 1.56	8.43 ± 1.78 *	P^1^ < 0.05	17.37 ± 4.50	P^2^ < 0.05
V/F (mg/kg)/(μg/mL)				53.51 ± 26.39	

AUMC, area under the first moment curve; MRT, mean residence time; Cl/F, clearance; V/F, tissue distribution; p^1^, compared between magnolol in the DMSO group and in formulation B by intraperitoneal administration; p^2^, compared between magnolol in the formulation B group by intraperitoneal and oral administration;* *p* < 0.05; ** *p* < 0.01.

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
