# Peer review of "CYP3A Excipient-Based Microemulsion Prolongs the Effect of Magnolol on Ischemia Stroke Rats"

_pharmaceutics, 2020, doi:10.3390/pharmaceutics12080737_

Round 1
Reviewer 1 Report
The quality of the manuscript has been significantly improved. However, some small corrections still needed:
The exact composition of ME needed
In the MCAO experiments the DMSO only control group is missing. Give some explanation for that.
2.9.1 The incubation temperature still missing
Table 1. What was the surfactant?
The MCAO is written as MACO in some places, need to be corrected.
English should be polished.
Author Response
Reviewer #1
The quality of the manuscript has been significantly improved. However, some small corrections still needed:
The exact composition of ME needed
Response: We appreciate your suggestions. The detail composition of ME (Surfactant: Oil: water = 7:1:2, while the mixing ratio of surfactant was Tween 80: PEG 400: 1,2-PG=1:1:1. The mixing ratio of oil was TG: OA=3:1) was listed on Table 1 (Line 222-223 on page 6).
In the MCAO experiments the DMSO only control group is missing. Give some explanation for that.
Response: We appreciate your suggestions. Our previous study (Chen JH et al. Toxicol Appl Pharmacol 2014, 279, 294-302) showed that the DMSO only treatment did not attenuate the brain damage in ischemic rats. Thus, in this study, we focus on the effect of magnolol in different delivery vehicles.
2.9.1 The incubation temperature still missing
Response: We appreciate your suggestions and have added the temperature (Line 191-191 on page 5).
Table 1. What was the surfactant?
Response: The surfactant was Tween-80 and the co-surfactant were PEG 400 and 1,2-PG. The mixing ratio was Tween 80: PEG 400: 1,2-PG=1:1:1.
The MCAO is written as MACO in some places, need to be corrected.
Response: We appreciate your suggestions and have corrected the errors (Line 301 and 364 on page 11 and 12).
English should be polished.
Response: We appreciate your suggestions and have rechecked the manuscript.

Reviewer 2 Report
The authors present a comparison of Mangnolol in DMSO and microencapsulation when administered ip into mice. The auhtors that they are at least equivalent and in some instances the microencapsulation is superior to DMSO in pharmacokinetic parameters. In the abstract only the result from one positive assay is mentioned and not that there was no pharmacological benefit from treatment with magnolol versus control (grip strength and EBST), so this needs to be clearly stated in the abstract and manuscript - these reagents only reduced visual infarct size and did not have a pharmacological effect on ischemic brain injury caused by occlusion. Second since this is the key component of a traditional medicine, what is the oral bioavailability of magnolol? Oral clearance was significanly improved in the microencapsulated version; however, there were no oral administration results shown. Since ip is not a method of administration in humans, I would suggest that the authors provide at least basic PK data for oral administration of magnolol. If sufficient exposure is demonstrated p.o. then that comparison on infarct size would be very interesting to the readers.
Author Response
Reviewer #2
The authors present a comparison of Mangnolol in DMSO and microencapsulation when administered ip into mice. The auhtors that they are at least equivalent and in some instances the microencapsulation is superior to DMSO in pharmacokinetic parameters. In the abstract only the result from one positive assay is mentioned and not that there was no pharmacological benefit from treatment with magnolol versus control (grip strength and EBST), so this needs to be clearly stated in the abstract and manuscript - these reagents only reduced visual infarct size and did not have a pharmacological effect on ischemic brain injury caused by occlusion.
Response: We appreciate your suggestions and have rewritten the abstract (Line 24-29 on page 1) and the conclusion (Line 349-352 on page 12).
Second since this is the key component of a traditional medicine, what is the oral bioavailability of magnolol? Oral clearance was significanly improved in the microencapsulated version; however, there were no oral administration results shown. Since ip is not a method of administration in humans, I would suggest that the authors provide at least basic PK data for oral administration of magnolol. If sufficient exposure is demonstrated p.o. then that comparison on infarct size would be very interesting to the readers.
Response: We appreciate your suggestions. The oral bioavailability of magnolol is about 4% (Lin SP et al., Planta Med 2011 77(16):1800-5). We also conducted the oral administration PK study. For safety and toxicity issues, rats were not orally administrated magnolol in DMSO (Our IACUC would also not approve such a study). We highlighted the results in Results paragraph- Pharmacokinetic parameters of magnolol loaded ME (Line 238-242 on page 7), and in Discussion (Line 316-318 on page 12).
Reviewer 3 Report
Hello Authors
Well written paper on ME delivery system. The manuscript covers important aspects of formulation development and physiological studies.
Here are some comments,
- Solubility of magnolol in water, DMSO should also be presented.
- Ternary phase diagram: Explain the reason why formulation A&B were chosen.
- Instead of room temp, mention the actual temperature.
- Droplet size, zeta potential, viscosity: Provide further details. How much sample size, what kind of viscometer (not the model, but the principal of its working), etc.
Author Response
Reviewer #3
Hello Authors
Well written paper on ME delivery system. The manuscript covers important aspects of formulation development and physiological studies.
Here are some comments,
- Solubility of magnolol in water, DMSO should also be presented.
Response: We appreciate your suggestions. The solubility of magnolol in water was 1.24 mg/L at 25 °C (National Center for Biotechnology Information. PubChem Database. Magnolol, CID=72300) and in DMSO was approximately 16 mg/ml (Magnolol Product Information, Cayman Chemical). We highlighted it in introduction (Line 52-55 on page 2).
- Ternary phase diagram: Explain the reason why formulation A&B were chosen.
Response: We appreciate your suggestions. The formulation A & B have relatively high water content and suitable viscosity for IP inject. Thus, we choose those two formulations for further study.
- Instead of room temp, mention the actual temperature.
Response: We appreciate your suggestions and have added the actual temperature 25 °C ± 1 °C in manuscript (Line 92, 97, 101, and 132 on page 3; Line 142 on page 4; Line 191 and 207 on page 5).
- Droplet size, zeta potential, viscosity: Provide further details. How much sample size, what kind of viscometer (not the model, but the principal of its working), etc.
Response: We appreciate your suggestions. The sample size for droplet size, zeta potential and viscosity study was three (Line 89 on page 2, Line 92-93 on page 3, and Line 222 on page 6). We use the rotational viscometer for measurement (Line 91 on page 3).

Round 2
Reviewer 2 Report
the authors have responded to the reviewer's suggestions adequately
This manuscript is a resubmission of an earlier submission. The following is a list of the peer review reports and author responses from that submission.
Round 1
Reviewer 1 Report
Dear author,
I have read the present manuscript with high interest. The concept is novel but there needs a major revisions before acceptance.
- Lack of characterization is evident..please do a TEM/SEM of the microemulsion.
- Stability of these nanoformulation are not discussed in details. Please show their stability in serum.
- Fig 4 needs scale bar.
- The release of magnolol needs to be presented as % release.
- How the survivability of the rats improved after controlled delivery?
Author Response
Reviewer #1
I have read the present manuscript with high interest. The concept is novel but there needs a major revisions before acceptance.
- Lack of characterization is evident.please do a TEM/SEM of the microemulsion.
Response: We appreciate your suggestions and have conducted the TEM study to characterize the microemulsion. The shape of formulation B showed various appearances under the observation by TEM. The particle diameter of formulation B observed by TEM is either smaller than or equal to the hydrodynamic diameter measured by Nanoparticle Analyzer. We highlighted the methods in Material and Methods paragraph 2.3.6 (Line 103-107 on page 3), the results in Results paragraph- Physicochemical properties of selected microemulsions (Line 198-201 on page 5), and in Discussion (Line 294-305 on page 11).
- Stability of these nanoformulation are not discussed in details. Please show their stability in serum.
Response: We appreciate your suggestions and have conducted the serum stability study (Zhou et al., Drug Deliv 2019, 26, 886-897). The serum stability study result also showed that formulation B slightly reduced its droplet size from 663.1 ± 74.4 nm at 1 h to 553.1 ± 84.4 nm at 6 h but reduced to 409.1 ± 73.3 nm (p<0.05, compared to 1 h) at 12 h. This result suggested the relative stability of formulation B during blood circulation for at least 6 h. We highlighted the methods in Material and Methods paragraph 2.3.5 (Line 98-101 on page 3), the results in Results paragraph- Physicochemical properties of selected microemulsions (Line 195-198 on page 5) , and in Discussion (Line 294-298 on page 11).
- Fig 4 needs scale bar.
Response: Thank you for your suggestions. We add the scale bar in Figure 7 panel a, b, and c (Line 258 on page 10).
- The release of magnolol needs to be presented as % release.
Response: Thank you for your suggestions. We add one column of % of maximum solubility in Table 2 (Line 214 on page 7).
- How the survivability of the rats improved after controlled delivery?
Response: Our current animal study focused on the acute effects after 24 h of magnolol administration. Therefore, both treatment groups’ animals were 100% survival. We will design a more severe brain stroke animal protocol and extend the observation time to compare the survival rates of different controlled delivery methods in our next study.

Reviewer 2 Report
The present research has been addressed to develop a CYP3A-excipient base microemulsion formulation to improve the solubility and bioavailability of magnolol to increase its neuroprotective effect against ischemic brain injury. The results shows that the microemulsion formulation do not improve the protective effect of magnolol on brain infarct size, however the drug concentration increases in the brain after 24h when it is administrated in the microemulsion. The authors conclude that increased availability of magnolol in the microemulsion leads to a superior brain protection compare with magnolol in DMSO.
Clearly this conclusion is not supported by the obtained results. In addition, there are some queries related to this paper.
The introduction is somehow excessively superficial. For example, the CYP3A should be described in more depth, particularly in relation to its ability to enhance antioxidant and neuroprotective efficacy of magnolol in the ischemia by down-regulating p38/ MAPK, CHOP pathway and despite the references cited.
The methodology is too bare, so it induces many doubts. For example: In relation to the surgical model of brain ischemia, has a doppler been used to control the blood flow and therefore to obtain homogeneous infarcts in all the experimental animals?. This point is not clear and neither the number of animals used to ascertain the therapeutic effect of the formulation; particularly, the sentence in the figure 3: “data are expressed as mean ± SD of three independent experiments”, does it mean that three animals from each group have been used to calculate the infarct size?.
Why a dose of 30mg/kg magnolol in DMSO or in ME? This point must be clarified.
In the results, although the determinations showed that, after 24h from the intraperitoneal administration of magnolol in the microemulsion, the drug reached high concentration in the brain than when it was disolved in DMSO, the physicochemical properties of the selected microemulsion (formulation B) showed an excessive dropped size (nearly 700 nm) as well as a Z-potential negative. These properties don’t seem too favorable to drive the magnolol to the brain parenchyma, even more considering that the formulation has to go through BBB. This question should be explain, particularly taking into account that any evidence of the penetration of the microemulsion in the brain are presented (for example labeling the microemulsion with fluorescence and visualizing in the brain using confocal microscopy).
To present SEM and TEM images of the microemulsion also would help to understand its properties (size, polydispersion …)
MRI image and/or behavioral tests would allow to assess more properly the neuroprotective effects of the formulation.
Minor questions: There is a mistake in the line 177. Mice should be rats (line 226).
Author Response
Reviewer #2
The present research has been addressed to develop a CYP3A-excipient base microemulsion formulation to improve the solubility and bioavailability of magnolol to increase its neuroprotective effect against ischemic brain injury. The results shows that the microemulsion formulation do not improve the protective effect of magnolol on brain infarct size, however the drug concentration increases in the brain after 24h when it is administrated in the microemulsion. The authors conclude that increased availability of magnolol in the microemulsion leads to a superior brain protection compare with magnolol in DMSO.
Clearly this conclusion is not supported by the obtained results. In addition, there are some queries related to this paper.
Response: Thank you for your comments. We revised the discussion and also the sentence as the increased availability of magnolol in ME which may leads to superior ischemic brain protection (Line 292-293 on page 11).
The introduction is somehow excessively superficial. For example, the CYP3A should be described in more depth, particularly in relation to its ability to enhance antioxidant and neuroprotective efficacy of magnolol in the ischemia by down-regulating p38/ MAPK, CHOP pathway and despite the references cited.
Response: Thank you for your suggestions. We revised the introduction paragraph to emphasized the information about the exciptients such as Tween-80 and and Cremophor RH40 inhibited the CYP3A activity (Bravo Gonzalez et al., Biopharm Drug Dispos 2004 25, 37-49; Mountfield et al., Int J Pharm 2000, 211, 89-92) as well as the above two potential excipients based in the self-microemulsiflying formulation to reduce the metabolism of CYP3A substrates (Rao et al., J Huazhong Univ Sci Technolog Med Sci 2010, 30, 562-568). (Line 44-48 on page 1 and 2)
The methodology is too bare, so it induces many doubts. For example: In relation to the surgical model of brain ischemia, has a doppler been used to control the blood flow and therefore to obtain homogeneous infarcts in all the experimental animals? This point is not clear and neither the number of animals used to ascertain the therapeutic effect of the formulation; particularly, the sentence in the figure 3: “data are expressed as mean ± SD of three independent experiments”, does it mean that three animals from each group have been used to calculate the infarct size?.
Response: Thank you for your suggestions. We revised the Material and Method paragraph 2.5. Blood and brain sample collection for pharmacokinetic study (Line 115-126 on page 3) and paragraph 2.8. Surgical procedures for inducing a reversible ischemic stroke (Line 143-167 on page 4).
To obtain homogeneous infarcts in all the experimental animals, we conducted the elevated body swing test. While the MCA occlusion induced ischemic stroke was successful damaged the left focal cerebral, a biased swing to the right side as a “C-shaped lateral bending body” would be observed immediately (Safakheil et al., J Mol Neurosci 2020, 70, 724-737; Ingberg et al., BMC Neurosci 2015, 16, 50). Also, three animals from each group have been used to calculate the brain infarct size.
Why a dose of 30mg/kg magnolol in DMSO or in ME? This point must be clarified.
Response: Our previous study (Chen et al., Toxicol Appl Pharmacol 2014, 279, 294-302) demonstrated that 30mg/kg magnolol in DMSO reduced the total infracted volume and then protected neurons against ischemia injury. In this study, we followed the dosage. We also revised the Material and Method paragraph 2.9. Magnolol administration for pharmacodynamic study to clarify the point (Line 173-175 on page 4).
In the results, although the determinations showed that, after 24h from the intraperitoneal administration of magnolol in the microemulsion, the drug reached high concentration in the brain than when it was disolved in DMSO, the physicochemical properties of the selected microemulsion (formulation B) showed an excessive dropped size (nearly 700 nm) as well as a Z-potential negative. These properties don’t seem too favorable to drive the magnolol to the brain parenchyma, even more considering that the formulation has to go through BBB. This question should be explain, particularly taking into account that any evidence of the penetration of the microemulsion in the brain are presented (for example labeling the microemulsion with fluorescence and visualizing in the brain using confocal microscopy).
Response: We appreciate your suggestions. The selected formulation B is a clear solution that is entrapped in oil/water/surfactants. Its droplet size and zeta potential are around 700 nm and −15 mV, respectively. Zeta potential is an essential index of stability of the suspension. The greater the absolute zeta potential, the higher the stability of colloidal dispersions (Elzoghby, A.O. J Control Release 2013, 172, 1075-1091). ation for at least 6 h. The value of electronic conductivity can define the nature of the microemulsion. A previous study reports that microemulsion with a conductivity ranging from 10.3 to 52.5 μS/cm can be defined as a bicontinuous microemulsion (Subongkot et al., Int J Nanomedicine 2017, 12, 5585-5599).The conductivity of formulation B is about 16 μS/cm and thus can be characterized as a bicontinuous microemulsion. Our brain distribution study showed that magnolol can be detected via HPLC-UV after 1h intraperitoneal injection of magnolol in ME treatment group. Also, our assay used power regression in the concentration range of 0.02-20 μg/mL for magnolol. The inter- and intra-day assay accuracy (% error) and precision (% CV) were between -6.7 and 4.2% and 0.7 and 5.6%, respectively, for magnolol at a concentration of 0.02 μg/mL. This should be the direct and reliable evident that ME can deliver magnolol pass through the BBB into brain tissue.
To present SEM and TEM images of the microemulsion also would help to understand its properties (size, polydispersion …)
Response: We appreciate your suggestions and have conducted the TEM study to characterize the microemulsion. The shape of formulation B showed various appearances under the observation by TEM. The particle diameter of formulation B observed by TEM is either smaller than or equal to the hydrodynamic diameter measured by Nanoparticle Analyzer. We highlighted the methods in Material and Methods paragraph 2.3.6 (Line 103-107 on page 3), the results in Results paragraph- Physicochemical properties of selected microemulsions (Line 198-201 on page 5), and in Discussion (Line 294-305 on page 11).
MRI image and/or behavioral tests would allow to assess more properly the neuroprotective effects of the formulation.
Response: We appreciate your suggestions and have conducted the behavioral tests including the elevated body swing test (EBST) and the grip strength assessment to assess the neuroprotective effects. We highlighted the methods in Material and Methods paragraph 2.8. (Line 143-167 on page 4), the results in Results paragraph- Brain distribution and effects of magnolol on an induced ischemic stroke (Line 228-238 on page 7 and 8), and in Discussion (Line 273-279 on page 11).
Minor questions: There is a mistake in the line 177. Mice should be rats (line 226).
Response: Thank you for your suggestions. We corrected the mistakes (Line 230 on page 8, and Line 310 on page 11).

Reviewer 3 Report
The authors described here the effect of microemulsification on the neuroprotective profile of magnolol in focal cerebral ischemia in rats.
Although the results are clearly presented some basic information missing and the complete reorganization of the manuscript is needed.
Neither in the title, nor in the introduction there is information about that the excipients added to magnolol for formation of microemulsion have CYP3A4 inhibitory effect. This information should be emphesized as early as possible not only at the end of the article.
Some methodological details should be added to the article:
blood collection method, collected blood volumes, number of experimental animals, number of animals/time point.
How did the authors check the success of MCA occlusion? Was there any blood flow measurement or other technique used?
Figure 2: The time scale should be changed, I suggest an axis break between 6 and 20 hours and should show the initial phase by 30 min, 60 min, 90 min etc. timepoints.
Table 3: Abbreviations should be explained (AUMC, MRT, Cl/F)
Figure 4: What is the difference between c and b panels?
n=?
Regarding the different sizes of brain sections it would be suggested to calculate the infarct area as a percentage of the area of the whole section.
More data needed about the CYP3A4 inhibitory effect of Tween 80. Is there any data about the similar effects of PEG400 and 1,2 PG?
Author Response
Reviewer #3
The authors described here the effect of microemulsification on the neuroprotective profile of magnolol in focal cerebral ischemia in rats.
Although the results are clearly presented some basic information missing and the complete reorganization of the manuscript is needed.
Neither in the title, nor in the introduction there is information about that the excipients added to magnolol for formation of microemulsion have CYP3A4 inhibitory effect. This information should be emphesized as early as possible not only at the end of the article.
Response: Thank you for your suggestions. We revised the introduction paragraph to emphasized the information about the exciptients such as Tween-80 and and Cremophor RH40 inhibited the CYP3A activity (Bravo Gonzalez et al., Biopharm Drug Dispos 2004 25, 37-49; Mountfield et al., Int J Pharm 2000, 211, 89-92) as well as the above two potential excipients based in the self-microemulsiflying formulation to reduce the metabolism of CYP3A substrates (Rao et al., J Huazhong Univ Sci Technolog Med Sci 2010, 30, 562-568). (Line 44-48 on page 1 and 2)
Some methodological details should be added to the article:
blood collection method, collected blood volumes, number of experimental animals, number of animals/time point.
Response: Thank you for your suggestions. We revised the blood collection method, volumes, number of experimental animals, and number of animals for each time point in paragraph 2.5. Blood and brain sample collection for pharmacokinetic study. (Line 115-126 on page 3)
How did the authors check the success of MCA occlusion? Was there any blood flow measurement or other technique used?
Response: We check the success of MCA occlusion via the elevated body swing test. While the MCA occlusion induced ischemic stroke was successful damaged the left focal cerebral, a biased swing to the right side as a “C-shaped lateral bending body” would be observed immediately (Safakheil et al., J Mol Neurosci 2020, 70, 724-737; Ingberg et al., BMC Neurosci 2015, 16, 50). We also add the method of the elevated body swing test in paragraph 2.8.1 Behavioral tests. (Line 155-159 on page 4)
Figure 2: The time scale should be changed, I suggest an axis break between 6 and 20 hours and should show the initial phase by 30 min, 60 min, 90 min etc. timepoints.
Response: Thank you for your suggestions. We changed the time scale of figure 4. (Line 223 on page 7)
Table 3: Abbreviations should be explained (AUMC, MRT, Cl/F)
Response: Thank you for your suggestions. We explained the abbreviations as following:
AUMC, Area under the first Moment Curve;
MRT, Mean Residence Time;
Cl/F, Clearance. (Line 227 on page 7)
Figure 4: What is the difference between c and b panels? n=?
Regarding the different sizes of brain sections it would be suggested to calculate the infarct area as a percentage of the area of the whole section.
Response: Thank you for your suggestions. We reformatted the infarct area as a percentage of the area of the whole brain section in Figure 7. The panel was represented the stroke group, b was represented the magnolol in DMSO treatment group, c was represented the magnolol in ME treatment group while d was represented the calculation of infarct area in percentage. The animal numbers (n) for each group was three. (Line 251-256 on page 9)
More data needed about the CYP3A4 inhibitory effect of Tween 80. Is there any data about the similar effects of PEG400 and 1,2 PG?
Response: Thank you for your suggestions. Our currently study focused on the effect of Tween 80 base on the previous study report from Rao et al 2010 (Rao et al., J Huazhong Univ Sci Technolog Med Sci 2010, 30, 562-568). We will set up a new study to clarify the effects of co-surfactant such as PEG400 and 1,2-PG on CYP3A4 in our next study.
Round 2
Reviewer 1 Report
The authors has done the necessary revisions and hence it is recommended to accept as it's present form.
Author Response
Reviewer #1
The authors has done the necessary revisions and hence it is recommended to accept as it's present form.
Response: We appreciate your suggestions.
Reviewer 2 Report
The authors really have improved the manuscript however there are still some important unresolved issues. Such is the case of the MCAO model; so, while it is true that the behavior test are unequivocal effects of the infarcts, only the use of a doppler may assure the homogeneity of the infarcts. In addition, as the number of animals per group are very scarce (n=3), data of the infarct measures on TTC sections are underpowered and these results are very important in the weight of the research.
Finally, it is not convincingly clarified, how 700nm nanoparticles of magnolol in a ME showing negative Z-potential may cross BBB. The HPLC probes cannot discriminate if the magnolol-ME is inside the vascular bed or in the cerebral parenchyma.
Author Response
Reviewer #2
The authors really have improved the manuscript however there are still some important unresolved issues.
Such is the case of the MCAO model; so, while it is true that the behavior test are unequivocal effects of the infarcts, only the use of a doppler may assure the homogeneity of the infarcts.
Response: We appreciate your suggestions. A previous study (Zhuoyo et al., Brain Res 2017, 1665, 88-94) demonstrated that the percentages of infarction volume were around 30 to 40 % in a 70% decrease in regional cerebral blood flow in the MCA territory monitoring by a laser-Doppler flowmetry. The behavior score including body symmetry, gait, and circling behavior showed a positive correlation with the percentages of infarction volume. Our TTC stain shows that about 35% of brain infarction size in stroke group and the behavioral studies also show that the C-shaped lateral bending body and the grip strength dramatically decreased more than 50% compared to the baseline after MCAO surgery in all animals. Our TTC staining and behavior results are consistent with the previous study. We rewrite and highlighted it in Discussion (Line 283-292 on page 11).
We agree that using the doppler may assure the homogeneity of the infarcts. We confirmed that the Taiwan Mouse Clinic (National Science Council, Taiwan) has the Vevo LAZR-X system (includes a Vevo 3100 Imaging System and a Class IV Laser). We will submit an application to the Taiwan Mouse Clinic to use the instrument in our next study to get the homogeneity of the infarcts.
In addition, as the number of animals per group are very scarce (n=3), data of the infarct measures on TTC sections are underpowered and these results are very important in the weight of the research.
Response: We appreciate your suggestions. The data of the measures on TTC sections is re-performed by one-way ANOVA followed by the Bonferroni post-hoc test via IBM SPSS statistics software Version 20 (Line 258 on page 9). The p-value for DMSO and ME compared to the stoke group is 0.027 and 0.012, respectively.
Finally, it is not convincingly clarified, how 700nm nanoparticles of magnolol in a ME showing negative Z-potential may cross BBB. The HPLC probes cannot discriminate if the magnolol-ME is inside the vascular bed or in the cerebral parenchyma.
Response: We appreciate your comments. The strategy of blood-to-brain drug delivery involves improving BBB permeability of the drug-carrier conjugate. Nanocarriers such as micelles, nanoemulsion, and microemulsion are promising carrier vehicles for direct drug transport across the intact BBB. Because of the nano size, the drug-carrier conjugate can quickly enter the brain capillary endothelial cells by endocytosis or transcytosis, for example (Li et al., J Drug Target 2017, 25, 17-28). In our TEM observation results, we also find the image of formulation B images with relatively small particle size (about 50nm, Figure S3). The small size particle of micelle might improve the magnolol delivery form blood to the brain. The conductivity of formulation B is about 16 μS/cm, it can be characterized as a bicontinuous microemulsion. The amphiphilic nature of the bicontinuous microemulsion can also simultaneously enhance the permeability of magnolol-ME to pass through the blood-brain barrier (Line 306-323 on page 11-12). Our TTC stain shows the decreased brain infracted percentage in both magnolol-DMSO and magnolol-ME groups (Firuge 7 section D, Line 237-240 on page 8). This histology evident may also indicate that magnolol-ME indeed distributes into the cerebral parenchyma to archive the protective effect.

Reviewer 3 Report
The manuscript was significantly improved
After adding the following citation to 2.8.1.2, I suggest the manuscript for publication:
- DOI: 10.1016/j.brainresbull.2005.08.018
Author Response
Reviewer #3
The manuscript was significantly improved
After adding the following citation to 2.8.1.2, I suggest the manuscript for publication:
DOI: 10.1016/j.brainresbull.2005.08.018
Response: We appreciate your suggestions. We cited this reference (#34, Erdo et al., Brain Res Bull 2006, 68, 269-276.) to 2.8.1.2 (Line 167 on page 4).